# Whole blood transcriptomics analysis of Indonesians reveals translocated and pathogenic microbiota in blood

Katalina Bobowik[1,☯,¤a], Muhamad Fachrul[1,2,☯], Chelzie Crenna Darusallam[3,¤b], Pradiptajati Kusuma [3], Herawati Sudoyo[3], Clarissa A. Febinia[3,¤c], Safarina G. Malik[3], Christine A. Wells[4], Irene Gallego Romero[2,5,6,*]

1 School of BioSciences, The University of Melbourne, Parkville, Victoria, Australia, 2 Human Genomics and Evolution, St. Vincent's Institute of Medical Research, Fitzroy, Victoria, Australia, 3 Genome Diversity and Diseases Division, Mochtar Riady Institute for Nanotechnology, Universitas Pelita Harapan, Tangerang, Banten, Indonesia, 4 Department of Anatomy and Physiology, Stem Cell Systems, School of Biomedical Sciences, Faculty of Medicine, Dentistry and Health Sciences, The University of Melbourne, Parkville, Victoria, Australia, 5 School of Medicine, The University of Melbourne, Parkville, Victoria, Australia, 6 Center for Genomics, Evolution and Medicine, Institute of Genomics, University of Tartu, Tartu, Estonia

¤a Current address: Burnet Institute, Melbourne, Victoria, Australia; The Australian National University, Canberra, Australian Capital Territory, Australia
¤b Current address: Genos Laboratory, Jakarta, Indonesia
¤c Current address: Department of Archaeology, University of Cambridge, Cambridge, United Kingdom
☯ These authors contributed equally to this work.
* irene.gallego@svi.edu.au

**Data availability statement:** All Indonesian data are available from the European Genome-phenome Archive (accession number

## Abstract

Pathogens found within local environments are a major cause of morbidity and mortality. This is particularly true in Indonesia, where infectious diseases such as malaria or dengue are a significant part of the disease burden. Unequal investment in medical funding throughout Indonesia, particularly in rural areas, has resulted in under-reporting of cases, making surveillance challenging. Here, we use transcriptome data from 117 healthy individuals living on the islands of Mentawai, Sumba, and the Indonesian side of New Guinea Island to explore which pathogens are present within whole blood. We identified diverse microbial taxa in RNA-sequencing data from whole blood but found no evidence of a consistent core microbiome across the Indonesian cohort. Yet, Flaviviridae and Plasmodium stood out as the most predominantly abundant taxa, particularly in samples from the easternmost island within our Indonesian dataset. The high prevalence of Plasmodium, the pathogen responsible for malaria, aligns with epidemiological data showing that the Indonesian part of New Guinea has the country's highest malaria rates. We also compare the Indonesian data to two other cohorts from Mali and UK and find a distinct microbiome profile for each group. Higher levels of dissimilarity were found between UK cohort (urban) compared to Indonesian and Malian cohorts (rural), where the former also have significantly lower within-population dissimilarity. This study provides a framework for RNA-seq as a possible retrospective surveillance tool and an insight to what makes up the transient human blood microbiome.

EGAS00001003671). Raw sequence reads for the Mali and UK cohorts were downloaded from SRA (accessions GSE52166 and GSE107991, respectively).

**Funding:** The author(s) received no specific funding for this work.

**Competing interests:** The authors have declared that no competing interests exist.

## Introduction

Pathogens are a major cause of morbidity and mortality, especially in the Global South [1–3]. Current knowledge of which taxa are present within remote regions of the world, along with how they impact health outcomes, remains limited. Not only is surveillance complex in these settings, but identifying which pathogens are responsible for disease symptoms can be challenging. For instance, although a pathogen may be identified in a population, it might not be the causative agent of disease due to indistinguishable symptoms and cross-reactivity of multiple pathogens [4]. Having a more detailed understanding of which pathogens are the major causes of morbidity across different global populations can focus elimination efforts on specific pathogens and aid in more targeted disease therapeutics.

Blood transcriptome data can be used to empirically test which blood-borne pathogens are present within an individual. Along with pathogenic organisms that infect blood cells, such as arthropod-borne pathogens [5,6] and various viruses [7,8], emerging research has shown that even bacteria and fungi can release DNA and RNA into blood [9]. For example, commensal bacteria [10,11], viruses [12,13], fungi [14], and archaea [15] have all been identified independently in multiple studies of human blood. While not yet common, the use of blood as a surveillance tool is growing. For instance, Kafetzopoulou et al. [16] used plasma samples from Lassa fever patients to identify the emergence of new strains, while two recent studies used whole blood samples from critically endangered mammals [17] and songbirds [18] to aid in the characterisation of diverse blood parasites.

Still, the topic of whether consistent microbial communities exist across healthy individuals remains highly debatable. Recent studies have found no evidence of a core microbiome circulating in the blood of healthy individuals [19]. This suggests that the blood microbiome is more transient in nature, comprising either commensal microbes translocated from other body sites or those involved in pathogenic activity and other disease states [20]. However, not enough studies have been done to confirm whether this lack of a core microbiome is also consistent in regions where infectious diseases are endemic.

Indonesia is a country with large numbers of endemic and emerging infectious diseases [21], making it a crucially important location to monitor and understand the effects of pathogens on human hosts. While several endemic diseases have been successfully reduced or eliminated in Indonesia [22], pathogen abundance can still be high in more rural areas, which tend to have less access to medical resources [22–24]. We have previously sampled individuals from three remote islands in Indonesia—Mentawai, Sumba, and the Indonesian side of New Guinea Island—and showed that individuals from the easternmost side of Indonesia (New Guinea Island) show widespread differences in immune gene expression levels compared to individuals from western (Mentawai) or central (Sumba) Indonesian islands [25]. While some of this variation is likely attributable to the different genetic ancestries of individuals in these islands [25,26], another significant contributor may be environmental differences, such as pathogenic load. Indeed, both *Plasmodium falciparum* and *Plasmodium vivax* are detectable at low levels within whole blood of some of these individuals [27], with a higher Plasmodium abundance within individuals from New Guinea Island. This observation suggests that pathogen loads are variable across the country, and that a non-targeted, transcriptomic approach can be used to capture these differences.

To characterise blood-borne microorganisms within Indonesia, this study utilises transcriptomic data collected from whole blood within these three previously described groups: the peoples of Mentawai and Sumba, and the Korowai. These populations span a gradient from west to east across Indonesia, thus capturing pathogens along the main geographical axis of the country. Unlike more populous regions within Indonesia, these three islands offer

a valuable model for understanding pathogen load in areas with limited resources, where reporting and traditional surveillance methods are often challenging. As such, they provide important insights from under-represented regions.

## Materials and methods

The Indonesian dataset consists of 101 base-pair, paired-end RNA-seq data from the whole blood of 117 healthy individuals living on the Indonesian islands of Sumba (n = 49), Mentawai (n = 48), and on the Indonesian side of New Guinea Island (n = 20, as described in [25]; all Indonesian data are available from the European Genome-phenome Archive study EGAS00001003671). All collections and analyses followed protocols for the protection of human subjects established by institutional review boards at the Eijkman Institute (EIREC #90 and #126); the analyses in this publication were additionally approved by University of Melbourne's Human Ethics Advisory Group (1851639.1). In the original Natri et al. study [25], additional 6 libraries were generated to serve as technical replicates between sequencing batches, however for our study we only retained the replicate with the highest read depth. Samples for the dataset were collected using Tempus Blood RNA Tubes (Applied Biosystems) and RNA-Seq libraries were prepared using Illumina's Globin-Zero Gold rRNA Removal Kit. Samples were then sequenced on an Illumina HiSeq 2500, resulting in an average read depth of 30 million read pairs per individual (S1–S4 Tables).

To compare the Indonesian dataset to other global populations, we searched for multiple publicly available transcriptomic datasets of whole blood from self-described healthy human donors. To control for technical covariates, we limited ourselves to datasets prepared using a globin depletion method and collected using Tempus Blood RNA Tubes, the same process followed by our own Indonesian dataset. We identified two publicly available datasets as controls. The first dataset comes from studies by Tran et al. [28,29], and consists of 101-bp human whole blood RNA-seq data, hereafter referred to as the Mali study. As described in [29], samples were collected from individuals living in the rural village of Kalifabougou, Mali, an area where there is a high rate of seasonal *P. falciparum* transmission. Raw sequence reads for this study were downloaded from SRA study GSE52166 and only samples which were collected pre-infection (n = 54) were used. The second dataset comes from Singhania et al. [30] consisting of 75-bp human whole blood RNA-seq data, collected from volunteers at the MRC National Institute for Medical Research in London, UK, hereafter referred to as the UK study. Raw sequence reads for this study were downloaded from SRA study GSE107991 and only healthy control samples (n = 12; all of European ethnicity) were used.

### RNA sequencing data processing

To investigate the metatranscriptome of whole blood, we put all reads through a stringent quality control pipeline. RNA-seq reads from all datasets went through an initial sample quality analysis using FastQC v. 0.11.5 [31]. To ensure reads were of high quality and free from artefacts, leading and trailing bases below a Phred quality score of 20 were removed and universal Illumina adapter sequences were trimmed (TruSeq3-PE.fa) using Trimmomatic v. 0.36 [32]. For comparisons between the Indonesian, Malian, and UK populations, the Malian and Indonesian datasets were trimmed to 75-bp, which is the read length of the UK dataset. We did this to control for differences in mappability and taxa identification associated with read length.

Paired-end RNA-seq reads were first aligned to the human genome (GRCh38, Ensembl release 90: August 2017) with STAR v. 2.5.3a [33] using the two-pass alignment mode and default parameters, and only reads that did not map to the human genome were retained for

further analysis. This step was performed to reduce the total library size to only pathogen candidates, and significantly decreases subsequent processing time. Unmapped sequencing reads were then processed using KneadData v. 0.7.4, which uses BMTagger [34] and Tandem Repeats Finder (TRF) [35] to remove human contaminant reads and tandem repeats, respectively. Using Kneaddata, BMtagger and TRF were run with default parameters. This resulted in a mean of 39,863 and 58,424 reads per sample for the 101-bp (S1 table) and 75-bp (S2 table) Indonesian datasets, respectively. For the 75-bp Malian (S3 table) and UK (S4 table) datasets, this resulted in a mean of 300,123 and 422,404 reads per sample, respectively.

## Mapping and metagenomic classification

Processed metagenomic reads were mapped using KMA v. 1.2.21 [36] against a filtered NCBI nt reference database, where artificial sequences and environmental sequences without valid taxonomic IDs were excluded [37] (downloaded on June 28, 2019 from https://researchdata.edu.au/indexed-reference-databases-kma-ccmetagen/1371207). We mapped paired-end reads using default settings and the following additional flags: -ef (extended features) was used to calculate reads as the total number of fragments, -1t1 was used for one read to one template (no splicing allowed in the reads), and -apm was set to $p$ which rewards pairing of reads. After mapping, we performed read classification using CCMetagen v. 1.2.2 [38] with default settings for paired-end reads. Read depth was calculated using the number of fragments with the read depth set to 1 so that we could analyse all possible matches. For the Indonesian dataset, these steps resulted in a mean of 6,480 reads per sample, which dropped to 4,579 when we trimmed reads to 75-bp (S2 table). For the 75-bp Malian (S3 table) and UK (S4 table) datasets, this resulted in a mean of 8,129 and 15,494 reads, respectively.

## Data filtering

After removing singletons to prevent spurious identification of taxa, we filter out reads mapped to the kingdoms Viridiplantae as these likely represented misassignments or poor quality annotation (S1 Fig, A–D) and further investigated the metazoan reads. We found that the majority of these mapped to the phylum Chordata (S1 Fig, E–H). We therefore decided to discard all reads mapping to Metazoa from subsequent analysis, as BLAST analysis of confirmed that these were reads that mapped equally well to the human genome. In addition, we also chose to remove taxa with no taxonomic rank assigned at the superkingdom level, as these taxa could not be linked to any known species. After removing Viridiplantae, Metazoa, and taxa with no taxonomic rank assigned at the superkingdom level, we obtained a mean of 905 reads in the Indonesian dataset (a mean of 694 for the 75-bp Indonesian reads; S2 table), 546 for the 75-bp Malian dataset (S3 table), and 5,230 for the 75-bp UK dataset (S4 table; S2 Fig).

## Sample clustering

To correct for uneven library depth between samples and the compositional nature of microbiome data [39], we applied a center log ratio (CLR) transformation [40] to the taxa abundance matrix when performing principal component analysis (PCA). Since a high number of zeros were present in the data, which CLR transformation is sensitive to [41], we chose to merge the abundance matrix at the phylum level. For this reason, we also performed analyses at the phylum level for all subsequent analyses utilising CLR-transformation.

Throughout, analyses are reported at the taxonomic level at which they were carried out, unless otherwise noted.

### Differential abundance testing and diversity estimation

We used ANOVA-like differential expression (ALDEx2) [42–44] to test for differences in species composition between populations, which applies CLR-transformation to correct for uneven library depth and data compositionality [43]. We performed differential abundance testing at the phylum level using the default Welch's t-test and default 128 Monte Carlo simulations. For alpha and beta diversity estimates, we used count abundances at the phylum level without removing singletons using the package DivNet v. 0.3.6 [45], which expects the presence of singletons in order to model species richness [45]. We used Wilcoxon rank-sum test to test significant differences between alpha and beta diversity estimates.

Code for all analyses is available at https://gitlab.svi.edu.au/igr-lab/indo_blood_microbiome.

## Results

### The blood microbiome of Indonesians

To provide a more comprehensive understanding of the blood microbiome of remote populations within Indonesia, we analysed unmapped reads from previously published whole blood transcriptomes, collected from 117 Indonesian individuals living on the islands of Mentawai (MTW) in western Indonesia, and Sumba (SMB) in central Indonesia, as well as the Korowai (KOR), a group living on the Indonesian side of New Guinea Island. The human samples have been extensively described [25,26]. After extensive quality control, we obtained a mean library size of 6,480 taxonomically informative reads after the removal of singletons (range: 2,212–48,471; S1 table). We assigned these reads to a total of 50 taxa across all phylogenetic levels, including 25 distinct taxa at the family level. As reads were predominantly assigned to Metazoan taxa, including *Homo sapiens*, further filtering was done; this resulted in an average of 3,923 reads across 27 samples that passed filtering, mapping to 15 distinct taxa at family level. *Plasmodiidae* (54.5% of the total read pool across all individuals) and *Flaviviridae* (40.8% of reads) were families with most reads assigned (Fig 1A). To control for sparsity in the abundance matrix, which is crucial when performing CLR-transformation [41], we also analysed the abundance of taxa at the phylum level in tests applying a CLR transformation to the data. Analysis of microbial reads at the phylum level resulted in the identification of 9 taxa, with Apicomplexa (54.3% of reads, within which 99.9% of reads mapped to the family *Plasmodiidae*), Kitrinoviricota (41% of reads, within which 100% of reads mapped to *Flaviviridae*), Ascomycota (1.6% of reads), and Pseudomonadota (0.8% of reads) making up the majority. These estimates of Apicomplexa load are higher than our previous estimates of Plasmodium burden [27], where we used a different, more conservative approach. We observed that the microbiome composition varied substantially between islands. In Korowai and Sumba populations, the majority of samples had reads assigned to either Apicomplexa (71.3% and 67.3% of reads) or Kitrinoviricota (28.1% and 26.6% of reads, respectively), whereas majority of reads in Mentawai samples mapped to Kitrinoviricota (91.8%).

PCA of the CLR-transformed taxonomic matrix showed sample clustering clearly driven by the phyla Apicomplexa (Fig 1B) and Kitrinoviricota (Fig 1C). We found that PC1, which captured over 40% of the variation, separated individuals by their abundance of either Apicomplexa (Pearson's r = 0.665) or Kitrinoviricota (Pearson's r = 0.917), as well as separating the Korowai from most of the populations of Mentawai and Sumba (Fig 1B and 1C). PC2 is

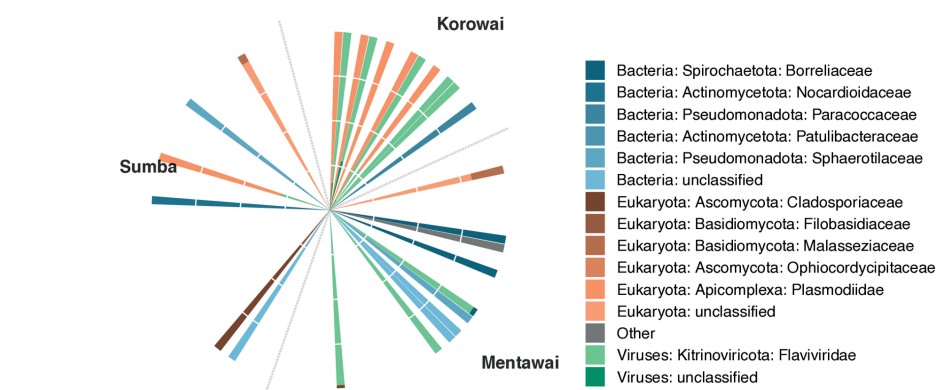

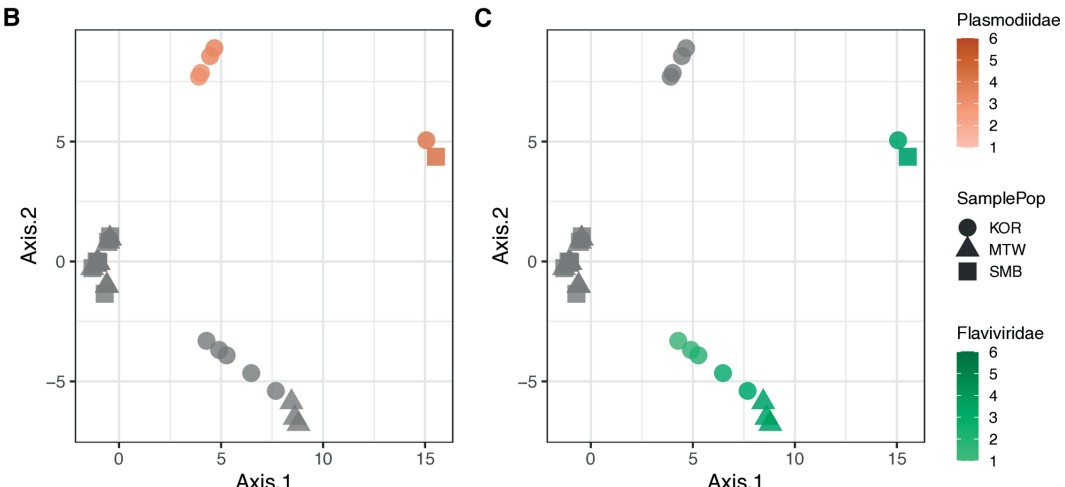

**Fig 1. The blood metatranscriptome of the Indonesian populations.** A) Circular barplot showing relative abundance (as percentage of reads) of the detected taxa within each individual in the Indonesian dataset, resolved at the family level. Bacteria are shown in blue, eukaryotes in orange, and viruses in green. KOR = Korowai; MTW = Mentawai; SMB = Sumba. Taxon labels include both phylum and family information. Empty bars represent individuals with no detected non-human RNA reads. B) Principal component analysis of the CLR-normalised taxa abundance data at the phylum level. Plotting shapes indicate population while $\log_{10}$ *Plasmodiidae* abundance is indicated in orange and C) green for *Flaviviridae*.

driven by the abundance of Apicomplexa (Pearson's r = 0.736) and could further be seen to separate between samples with a high abundance of Apicomplexa and Kitrinoviricota (Fig 1B and 1C). No clustering based on batch effect was seen from the first 8 PCs (S3 Fig).

## Microbiome diversity between island populations

As we are interested in whether there are observable differences in blood microbiomes between Indonesian island populations, we next performed differential abundance testing between the three groups using the ALDEx2 package [42–44]. Despite Apicomplexa having the largest abundance differences between sites, differential abundance testing at the phylum

level did not result in significant differences in between islands, either before or after the BH adjustment (Fig 2A, 2B, and 2C).

The diversity and types of microbes within human tissues can be an indicator of the overall health of an individual, and of a population [11,46]. We therefore analysed levels of alpha (within individual) and beta (between individual) diversity within the three islands using DivNet [45], again at the phylum level. We found that while alpha diversity estimates were overall comparably lower in individuals from Korowai and Mentawai than in

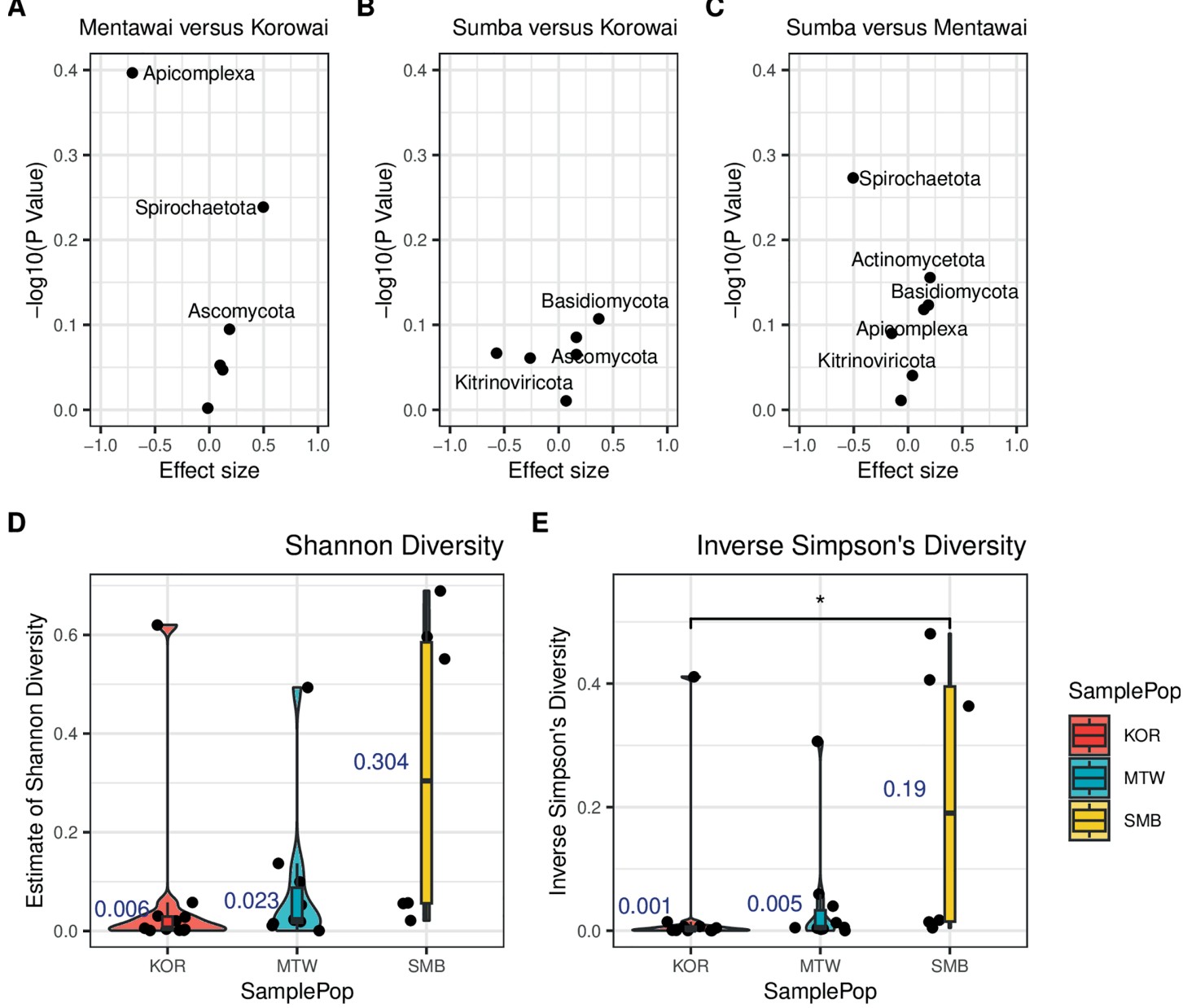

**Fig 2. Blood microbiomes are not statistically different between island populations.** A) Volcano plot of BH-adjusted p-values from Welch's t-test and the effect size for each taxa at the phylum level, in Mentawai versus Korowai B) Sumba versus Korowai and C) Sumba versus Mentawai. D) Estimates of Shannon and E) inverse Simpson diversity within each population (median in blue text). KOR = Korowai; MTW = Mentawai; SMB = Sumba.

individuals from Sumba, they were still lowest in individuals from the Korowai population. This was true for both estimates of Shannon diversity (median Shannon KOR = 0.006; MTW = 0.023; SMB = 0.304; Fig 2D) and inverse Simpson diversity indices (mean inverse Simpson KOR = 0.001; MTW = 0.005; SMB = 0.19; Fig 2E), though significant difference was only found between the inverse Simpson diversity indices between Korowai and Sumba (FDR adjusted Wilcoxon rank-sum test p = 0.044). This observation was likely driven by the high abundance of Apicomplexa reads amongst the Korowai, which account for the majority of the available read pool in these individuals, and therefore drive overall diversity rates down. We found that comparisons between populations resulted in similarly high estimates of Bray-Curtis dissimilarity with no significant differences (S4 Fig), which again mostly reflects the sparsity of the dataset, even between samples of the same island group.

## Microbiomes are distinct between global populations

To test whether blood microbiomes in Indonesia differ from those of other global populations, we also analysed microbiome data from two other publicly available datasets of whole blood transcriptomes. This includes 54 healthy individuals living in Kalifabougou, Mali [28,29], which represents the microbiome of individuals living in rural environments, and 12 healthy individuals collected from the city of London in the United Kingdom [30], representing the blood microbiome of individuals living in a highly urbanised environment. Similar to our Indonesian datasets, Kalifabougou is a malaria-endemic region and the majority of residents engage in subsistence farming practices [47].

After processing of reads as above, we obtained a mean library size of 15,494 reads (range: 4,493 - 33,711) for the UK dataset (S4 table) and 8,129 (range: 1,637 - 180,484) for the Malian dataset (S3 table) after the removal of singletons, respectively. This difference in depths is attributable to different numbers of reads being filtered out at different processing stages in the three datasets, as all three had similar starting read depths. All datasets lost significant numbers of reads when we filter reads assigned to either Viridiplantae or Metazoa (S1–S4 tables; S2 Fig). In the UK dataset, we identified a total of 101 distinct taxa across all phylogenetic levels. The majority of reads assigned to the bacterial phylum Pseudomonadota (81.2% of the total read pool across all individuals) and the fungal phylum Basidiomycota (11.2% of reads; Fig 3). Within the Malian dataset, we found 41 distinct taxa across all phylogenetic levels, the majority of which were Actinomycetota (41.2% of reads), followed by Artverviricota (18.4% of reads), Apicomplexa (10.7% of reads), Kitrinoviricota (9.6% of reads), Bacillota (5.9% of reads), and Ascomycota (4.2% of reads; Fig 3). Although there is a substantial difference in read depths between all three data sets, saturation curves show systematic similarity in diversity between the Indonesian and Mali samples (S5 Fig).

We performed differential abundance testing between the Indonesian, Malian, and UK datasets. Only Actinomycetota (FDR adjusted Welch's t-test p = 0.026) was found to be significantly differentially abundant between Malian and Indonesian individuals (Fig 4A; S5 table). Kitrinoviricota was found to be significantly differentially abundant prior to FDR correction (Welch's t-test p = 0.011; S5 table). When comparing blood microbiomes between the UK and Indonesian populations, we found 2 differentially abundant phyla, the most significant being Pseudomonadota and Kitrinoviricota, the former more abundant in the UK population and the latter in the Indonesian population (FDR adjusted Welch's t-test p = 3.76 $\times$ $10^{-9}$ and 0.02, respectively; Fig 4B; S6 table).

We next repeated differential abundance testing using only the Korowai as the Indonesian comparison group due to them containing the most pathogenic reads. We found that the comparisons yielded very similar results, with only Actinomycetota being significantly

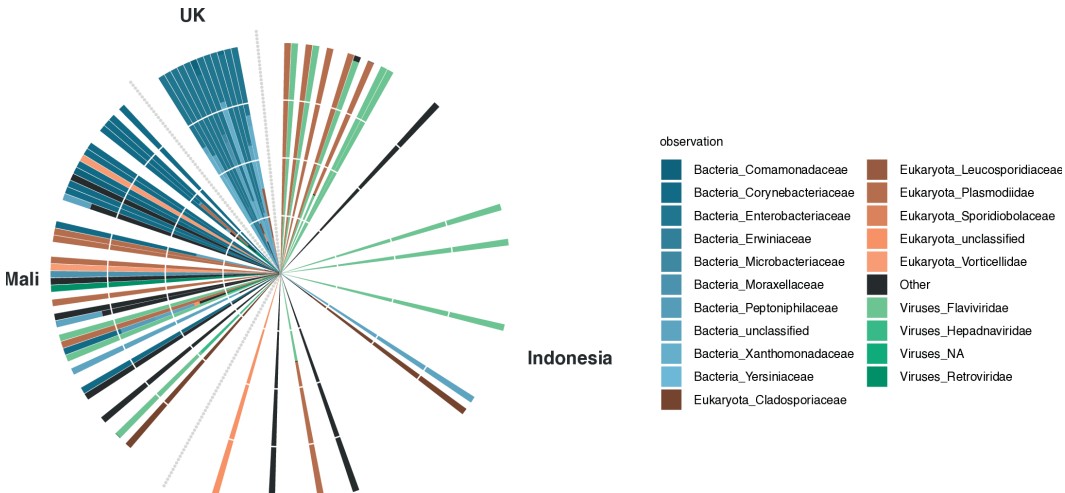

**Fig 3. Relative abundance of the top 20 taxa within the Indonesian, Malian, and UK dataset at the superkingdom, phylum, and family level.** Bacteria are shown in blue, eukaryotes in orange, and viruses in green.

differentially abundant between Mali and Korowai samples (FDR adjusted Welch's t-test p = 0.036; S7 table) and Pseudomonadota between UK and Korowai samples (FDR adjusted Welch's t-test p = $3.2 \times 10^{-6}$; S6 Fig; S8 table).

To identify overall trends between whole blood microbiomes of Indonesians and that of other populations, we next performed PCA on the CLR-transformed abundance matrix containing the Indonesian, UK, and Malian samples. Microbiomes clearly differed between countries as shown in PCs 1-2, yielding a separate cluster for each dataset (Fig 4C); meanwhile, PCs 3 and 4 did not show any clear clustering of the populations and instead were driven by *Plasmodiidae* (Fig 4D) and *Flaviviridae* loads (Fig 4E). The separation of UK samples, which can be seen in PC2, is recapitulated by the Bray-Curtis distance estimates where between-population comparisons with the UK cohort showed greater dissimilarities and within-UK estimate having the lowest within-population estimate (Fig 4F).

Finally, to understand species richness in blood microbiomes between populations, we again analysed levels of alpha diversity in each of the three global datasets. We found that the UK samples had the lowest Shannon (mean Shannon = 0.195; S7 FigA) and inverse Simpson diversity values (mean inverse Simpson = 0.098; S7 Fig, B), followed by individuals from Mali (mean Shannon = 0.208, mean inverse Simpson = 0.11), then Indonesia (mean Shannon = 0.27, mean inverse Simpson = 0.116). We also note that the UK population has the highest sequencing depth out of the three populations (S4 table) and consequently the greatest power to detect rare taxa, and therefore these estimates likely reflect true rates of lower diversity within the UK population.

## Discussion

Our understanding of pathogens found within remote regions of Indonesia, along with their impact on gene expression, is limited. Here, we have investigated to what extent microbial taxa can be detected within whole blood, and whether a core blood microbiome could be profiled. We did not detect taxa that constitute a core Indonesian whole blood microbiome, yet found evidence of strong pathogenic signals. This is consistent with recent findings of how the blood of healthy individuals do not support a consistent core microbial community [19].

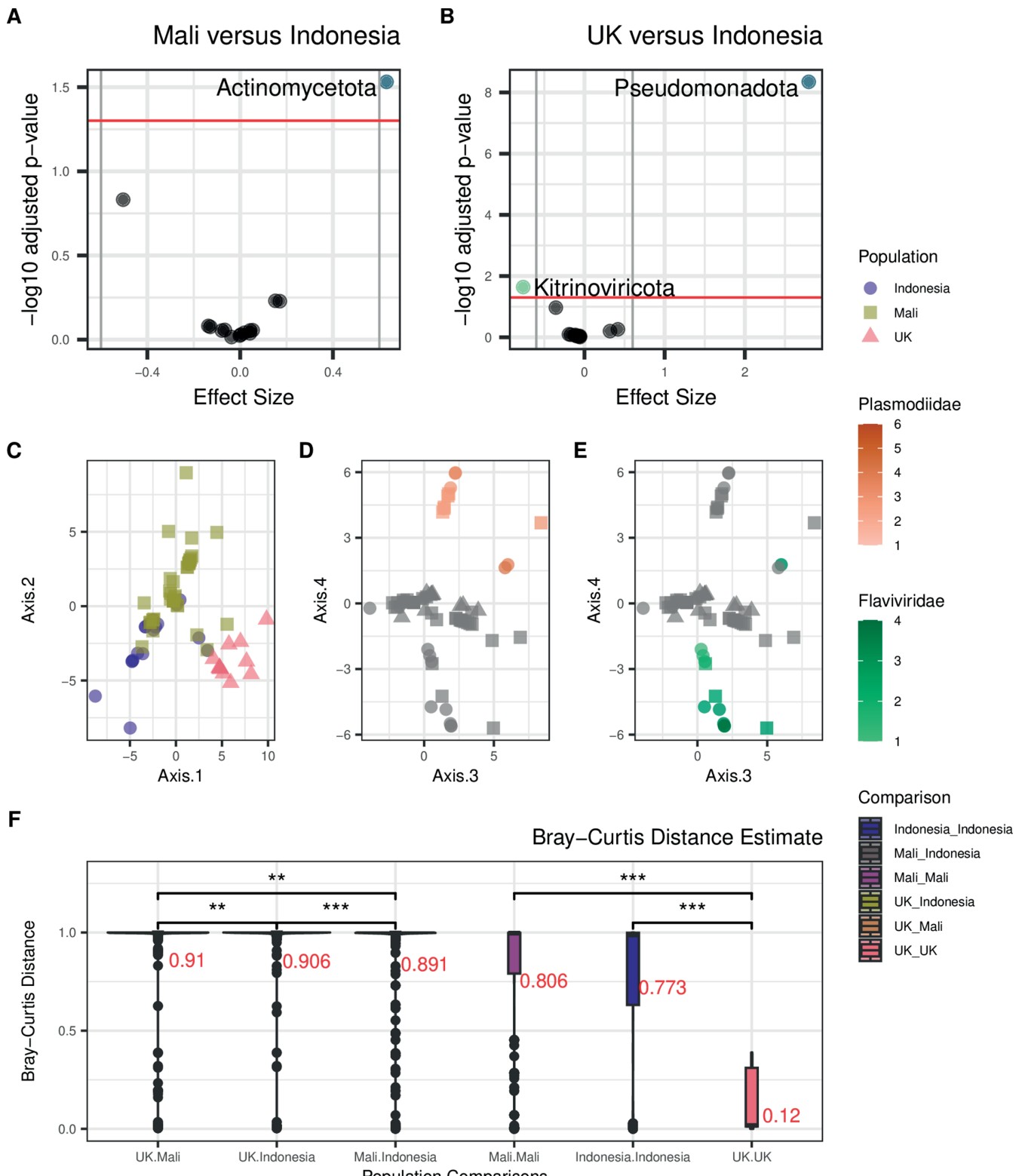

**Fig 4. Taxa differences between Indonesian individuals and other global populations.** A) Volcano plot of BH adjusted p-values from Welch's t-test for each phyla in Malian versus Indonesian individuals and B) UK versus Indonesian individuals. Taxa with an FDR-corrected p-value below 0.05 (above red threshold line) are coloured by superkingdom (blue: bacteria; green: viruses). C) Principal components (PCs) 1 and 2 of the CLR-normalised taxa abundance data at the phylum level, colored by population. D) PCs 3 and 4 of the same data colored by *Plasmodiidae* and E) *Flaviviridae* loads. F) Bray-Curtis distance estimates for Indonesian, Malian, and UK population comparisons at the phylum level (mean in red text).

We found evidence for the presence of eukaryotes and viruses, all of which have previously been characterised in blood transcriptomes [48]. This study supports a growing body of research suggesting that rather than being a sterile environment, a variety of taxa reside transiently within whole blood, and understanding their occurrence may facilitate better understanding of diseases and conditions in different populations.

Despite our attempts to remove contaminant human reads and tandem repeats prior to classification, we still identified significant numbers of reads that mapped to *Homo sapiens* and had to filter out reads that mapped to Metazoa. This reflects the issue raised by a recent study where Gihawi et al. reanalyzed a large-scale tumor microbiome study and found most bacterial reads to be misclassified human reads, largely due to the human genome being excluded from the classification reference database [49]. They also highlighted how the inclusion of draft bacterial genomes, which are often contaminated with human reads, contributed to the overestimation of bacterial species. Our results serve as another example of the importance of careful quality control measures to minimize false assignment of microbial reads, particularly by including the human genome and when possible using only complete microbial genomes during the classification process.

Despite not finding a core whole blood microbiome in the population, we identified two phyla that were dominant in multiple samples, namely Apicomplexa (driven nearly exclusively by the family Plasmodiidae) and Kitrinoviricota (driven by the family Flaviviridae). From taxonomic profiling, we could attribute Kitrinoviricota viral signals to the family Flaviviridae, which is a family of viruses primarily found in mosquitos and ticks, and is responsible for multiple human illnesses including Dengue in Indonesia [50–52]. Around 3.6% of the reads could be further specified as belonging to the *Pegivirus* genus, yet we were unable to refine this assignment for the majority of the reads. The *Pegivirus* genus includes the human pegivirus (HPgV-1), a non-cytopathic lymphotropic virus previously associated with increased potential risk of lymphoma and reduction of disease progression caused by HIV-1 during co-infection [53]. For Apicomplexa, we could attribute 99.9% of reads to the family Plasmodiidae, of which *Plasmodium falciparum* and *Plasmodium vivax* are endemic throughout Indonesia [54].

Of all the Indonesian island populations in this study, we found that the Korowai had the highest abundance of the two pathogens. The Indonesian side of New Guinea Island is documented to have the highest rates of malaria in Indonesia, contributing up to 94% of all national cases [55–57], as well as the lowest number of healthcare facilities [58]; our results corroborate existing observations of a high endemic pathogen load within this region.

We also profiled and compared the blood microbiome of Malian and UK populations to the Indonesian samples. Bray-Curtis distance estimates showed that the Indonesian, Malian, and UK populations had high levels of dissimilarity from one another (Fig 4F). We also found differences in diversity between Indonesian and Malian populations (rural) compared to the UK (urban). Beta diversity estimates were higher in Malian and Indonesian populations, although the UK population had the highest read depth out of all three populations; diversity in the UK samples was driven primarily by bacterial taxa whereas the other two sites were characterised by widespread presence of pathogen-derived reads. Previous studies have reported similar findings when it comes to diversity between rural and urban populations: the Hadza, a small hunter gatherer group in Tanzania was found to have more diverse gut microbiomes than Italian urban controls [59]. Another study comparing gut microbiomes of rural and urban environments found that urban microbiomes were distinct, and that urbanisation led to a loss of certain bacterial taxa [60].

Our findings are limited by the fact that all three datasets we considered were generated by different groups in different places, where biological variations might be affected by

differences during sampling and processing. Although our total sample sizes for the Indonesian samples are high—which is rare in studies of underrepresented populations, or more broadly, populations outside an urban, "western" environment—our total read depth is low, limiting the taxa we can detect in the population. Indeed, out of all three global populations, the Indonesian dataset had the lowest read depth (S5 Fig). However, in opportunistic studies such as this, meeting the conditions required for high sequencing depth is rare; sequencing depth of unmapped reads is sensitive to multiple factors, including sequencing platform, sample collection and processing strategy, and only two publicly available datasets that we could find met the requirements needed to withstand total microbiome depletion.

Mounting evidence suggests that some microorganisms are common inhabitants of whole blood yet are likely originating from the gut and oral cavities [61,62], as well as representing leakage from other parts of the body. We found stronger evidence of this in our analyses of the Malian and UK datasets. Taxa of the Actinomycetota phylum were found to be the most abundant in the Malian cohort, and around 84.6% of which could be further specified as *Corynebacterium tuberculostearicum*: a bacterium commonly found on human skin that is generally harmless, yet may play a role in skin health and disease [63,64]. In the UK cohort Pseudomonadota was found to be the most abundant, up to 51.3% of which could be further specified as Enterobacteriaceae, a bacterial family that encompasses species commonly found in the human gut such as *E. coli* and linked to inflammatory bowel disease [65]. Additionally, up to 25% of the Pseudomonadota reads were also defined as part of Xanthomonadaceae, a bacterial family that has been previously reported to colonize various hosts including the human skin [66,67]. Interestingly, the water-borne Xanthomonadaceae has also been reported as contaminants in DNA extraction kits and reagents ("kitomes"), including in a study identifying the placental microbiome [68].This further challenges the notion of a core human blood microbiome; our findings reaffirm how the blood microbiome is comprised of transient microbiota originating from other body sites and/or from pathogenic infections, and how, as a diagnostic medium, it may be hindered by limitations and variations of technical aspects.

A better understanding of which pathogens affect remote populations is crucial. Whole blood is one of the most abundant tissue types in RNA-seq analysis due to its relative ease of collection [69], and therefore its ability to provide information on environmental factors influencing disease phenotypes is ripe for investigation. In Indonesia, this is particularly important; Indonesia has a growing number of emerging infections [2,21], however proper surveillance in rural areas remains limited. Our study demonstrates the use of whole blood RNA for microbiome-based diagnostic purposes that perhaps may be more suited in a retrospective context. Profiling microbiome from whole blood RNA may not be the most efficient approach as a first-line diagnostic tool due to the time-intensive process it requires. Nevertheless, this study provides valuable retrospective surveillance information on blood-borne microorganisms within the region, which is a valuable step in understanding and eventually limiting the spread of endemic and emerging diseases. Extra care should be taken to understand the influences on both environmental and technical factors while using such an approach for pathogen detection.

## Supporting information

**S1 Fig. Summary of reads mapping to filtered taxa for the Indonesian (101BP and trimmed 75BP), Malian (75BP), and UK (75BP) populations.** A–D) Reads mapping to the Viridiplantae E–H) and Metazoa.
(TIF)

**S2 Fig. Read depth per individual library across all filtering steps.**
(TIF)

**S3 Fig. Principal component analysis of the CLR-normalised taxa abundance data at the phylum level from the Indonesian samples, colored by batches.** No clear clustering and strong Spearman correlation (r) were seen between batch and A) PCs 1-2, B) PCs 3-4, C) PCs 5-6, and D) PCs 7-8.
(TIF)

**S4 Fig. Bray-Curtis distance estimates for each island comparison at the phylum level.**
(TIF)

**S5 Fig. Rarefaction curves of species saturation per individual at varying read depths for the Indonesian, Malian, and UK populations.**
(TIF)

**S6 Fig. Taxa differences between samples from Korowai and other global populations.** A) Volcano plot of BH adjusted p-values from Welch's t-test for each phyla in the Korowai versus Malian populations and B) Korowai versus UK populations. Taxa with a BH-corrected p-value below 0.05 for are coloured by superkingdom (blue: bacteria).
(TIF)

**S7 Fig. Violin plots of A) Shannon diversity and B) inverse Simpson diversity for each population.**
(TIF)

**S1 Table. Read depth of each individual in the Indonesian dataset (101BP) after each filtering step.**
(XLSX)

**S2 Table. Read depth of each individual in the Indonesian dataset (75BP) after each filtering step.**
(XLSX)

**S3 Table. Read depth of each individual in the Malian dataset (75BP) after each filtering step.**
(XLSX)

**S4 Table. Read depth of each individual in the UK dataset (75BP) after each filtering step.**
(XLSX)

**S5 Table. Differential abundance analysis results (Welch's t-test BH-adjusted p = 0.05) at the phylum level between Malian and Indonesian datasets.**
(XLSX)

**S6 Table. Differential abundance analysis results (Welch's t-test BH-adjusted p = 0.05) at the phylum level between UK and Indonesian datasets.**
(XLSX)

**S7 Table. Differential abundance analysis results (Welch's t-test BH-adjusted p = 0.05) at the phylum level between Malian and Korowai samples.**
(XLSX)

**S8 Table. Differential abundance analysis results (Welch's t-test BH-adjusted p = 0.05) at the phylum level between UK and Korowai samples.**
(XLSX)

## Acknowledgments

We would like to acknowledge all of the study participants who generously consented to genome sequencing in the original study, as well as Emily R. Davenport, Murray P. Cox and members of the Gallego Romero group for helpful comments on the manuscript. St Vincent's Institute acknowledges the infrastructure support it receives from the National Health and Medical Research Council Independent Research Institutes Infrastructure Support Program and from the Victorian Government through its Operational Infrastructure Support Program. PK is supported by the Wellcome Trust International Training Fellowship (no. 222992/Z/21/Z).

## Author contributions

**Conceptualization:** Christine A. Wells, Irene Gallego Romero.

**Formal analysis:** Katalina Bobowik, Muhamad Fachrul.

**Investigation:** Katalina Bobowik.

**Methodology:** Katalina Bobowik, Clarissa A. Febinia.

**Project administration:** Irene Gallego Romero.

**Resources:** Chelzie Crenna Darusallam, Pradiptajati Kusuma, Herawati Sudoyo, Safarina G. Malik.

**Supervision:** Christine A. Wells, Irene Gallego Romero.

**Validation:** Muhamad Fachrul, Pradiptajati Kusuma, Clarissa A. Febinia.

**Visualization:** Katalina Bobowik, Muhamad Fachrul.

**Writing – original draft:** Katalina Bobowik, Irene Gallego Romero.

**Writing – review & editing:** Muhamad Fachrul, Chelzie Crenna Darusallam, Pradiptajati Kusuma, Herawati Sudoyo, Clarissa A. Febinia, Safarina G. Malik, Christine A. Wells, Irene Gallego Romero.

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
