## [Decision Letter · Decision Letter 0]

29 Oct 2024

PONE-D-24-24494The whole blood microbiome of Indonesians reveals translocated and pathogenic microbiotaPLOS ONE

Dear Dr. Irene Gallego Romero

Thank you for submitting your manuscript to PLOS ONE. After careful consideration, we feel that it has merit but does not fully meet PLOS ONE’s publication criteria as it currently stands. Therefore, we invite you to submit a revised version of the manuscript that addresses the points raised during the review process.

We look forward to receiving your revised manuscript.

Kind regards,

Rajesh Singh Rathore

Academic Editor

PLOS ONE

Journal Requirements:

1. When submitting your revision, we need you to address these additional requirements. Please ensure that your manuscript meets PLOS ONE's style requirements, including those for file naming. The PLOS ONE style templates can be found at https://journals.plos.org/plosone/s/file?id=wjVg/PLOSOne_formatting_sample_main_body.pdf and https://journals.plos.org/plosone/s/file?id=ba62/PLOSOne_formatting_sample_title_authors_affiliations.pdf 2. Please update your submission to use the PLOS LaTeX template. The template and more information on our requirements for LaTeX submissions can be found at http://journals.plos.org/plosone/s/latex. 3. Thank you for stating the following in the Acknowledgments Section of your manuscript: “We would like to acknowledge all of the study participants who generously consented to genome sequencing in the original study, as well as Emily R. Davenport, Murray P. Cox and members of the Gallego Romerogroup for helpful comments on the manuscript. St Vincent’s Institute acknowledges the infrastructuresupport it receives from the National Health and Medical Research Council Independent Research InstitutesInfrastructure Support Program and from the Victorian Government through its Operational Infrastruc ture Support Program. PK is supported by the Wellcome Trust International Training Fellowship (no.222992/Z/21/Z)” We note that you have provided funding information that is not currently declared in your Funding Statement. However, funding information should not appear in the Acknowledgments section or other areas of your manuscript. We will only publish funding information present in the Funding Statement section of the online submission form.  Please remove any funding-related text from the manuscript and let us know how you would like to update your Funding Statement. Currently, your Funding Statement reads as follows: “The authors received no specific funding for this work.” Please include your amended statements within your cover letter; we will change the online submission form on your behalf. 4. We notice that your supplementary figures are included in the manuscript file. Please remove them and upload them with the file type 'Supporting Information'. Please ensure that each Supporting Information file has a legend listed in the manuscript after the references list. 5. Please upload a copy of Supporting Information Figure/Table/etc. Supplementary tables 1 to 8  which you refer to in your text on pages 26 and 27.

Additional Editor Comments:

The article titled "The Whole Blood Microbiome of Indonesians Reveals Translocated and Pathogenic Microbiota" describes the use of transcriptomics data to investigate the presence of pathogens in whole blood samples specifically in Indonesian region.

The authors have addressed prior reviewer enquiries; still, major minor remarks are included as per other reviewer and some are as below:

a) Please explain how the author relates this study's impact to John P. Hanley et al. (2021) and Louis et al. (2005). What is the significance of whole blood transcriptome analysis, which identified distinct gene sets that correlated with viremia and revealed varying gene expression patterns before and after infection, implying immune state changes?

b) What are the hematological changes and cellular composition changes and Gene Expression Patterns in relational to host and pathogen?

c) Why did the author choose an RNA-based investigation while prior research using mNGS and shotgun metagenomics demonstrated a more extensive analysis?

d) Author mentioned “Recent studies have found no evidence of a core microbiome circulating in the blood of healthy Stanley plot individuals” and “However, not enough studies have been done to confirm whether this lack of a core microbiome is also consistent in regions where infectious diseases are endemic” but plethora of literature is available depicting role of core microbiome circulating in the blood of healthy individuals –

e) Study by Païssé et al. (2016), Study by Castillo et al. (2019), Blauwkamp et al….. so on, Nature study by Ling et al. (2023) - large-scale study analyzed blood samples from 9,770 healthy individuals.

f) Why does the author focus exclusively on Indonesia—specifically Mentawai, Sumba, and the Indonesian portion of New Guinea Island—as the data set for this study?

g) Enhance the methodology section by providing a detailed description that emphasizes sample collection, RNA isolation procedures, library preparation, quantification, and host DNA depletion techniques. Include the catalogue information for the kits used to facilitate replication by others, as well as the parameters employed for the Illumina HiSeq 2500, and ensure appropriate citations are included.

h) Provide the consent information for all human individuals (Weight, Hight, BMI etc).

i) Provide the mapped and unmapped paired-end data separately?

j) Please verify that the given data files encompass all individual samples in the repository.

k) Instead of a circular bar plot, the author should present a stanley plot that shows the interconnected changes between the samples; this will provide a better understanding of the data.

l) For compositional data analysis, the author must execute isometric log-ratio (ILR) transformation, which preserves all metric aspects of the data, coordinates with a non-singular covariance matrix, and allows for simpler interpretation of results. Alternatively, utilise the Robust Centred Log-Ratio (rCLR) and Inter-Quartile Log-Ratio (IQLR) Transformations.

m) The justification for doing Welch's t-test in this investigation is unclear author should use Limma-Voom, NOIseq, SAMseq, etc. for analysis?

n) In discussion part author mentioned “Our understanding of pathogens found within remote regions of Indonesia, along with their impact on gene expression, is limited. But several published studies on global scenario showed impact of pathogens are associated with environmental factors like temperature, humidity, location of longitude Natri et al., 2020 etc. As author used European data in this study but some studies mentioned the commonalities in gene regulation between Indonesia and Europe (Natri wt al., 2022), So, instead of using European data, I strongly recommend that the author conduct a comparative and conclusive study using data from other continents.

Reviewers' comments:

Reviewer's Responses to Questions

**Comments to the Author**

1. Is the manuscript technically sound, and do the data support the conclusions?

Reviewer #1: Yes

Reviewer #2: Yes

2. Has the statistical analysis been performed appropriately and rigorously? 

Reviewer #1: Yes

Reviewer #2: Yes

3. Have the authors made all data underlying the findings in their manuscript fully available?

Reviewer #1: Yes

Reviewer #2: Yes

4. Is the manuscript presented in an intelligible fashion and written in standard English?

Reviewer #1: Yes

Reviewer #2: Yes

5. Review Comments to the Author

Reviewer #1: The manuscript titled “The Whole Blood Microbiome of Indonesians Reveals Translocated and Pathogenic Microbiota” presents the use of transcriptomics data to explore the presence of pathogens in whole blood samples. The authors have responded to previous reviewer questions; however, a few minor comments are provided below:

1. If the authors agree, the title can be revised to "Whole Blood Transcriptomics Analysis of Indonesians…" to provide a clearer picture of the study.

2. If the primary aim of the study is to detect pathogens, a microbial-targeted metagenomic study could be considered. This approach may help address the issue of contaminated human reads that the authors encountered. Are there any previous studies that used metagenomic analysis for pathogen detection, and did they produce similar or different results? If using RNA-seq is more critical than metagenomics in this context, the authors could clarify this in the introduction or discussion section.

Reviewer #2: The manuscript under review focuses on a relevant but controversial topic: Assessing the state of the blood microbiome. I would like to focus on the ambiguity of the task that the authors set in their study. On the one hand, the presence of genome fragments of certain types of microorganisms in the blood was confirmed by experimental studies. On the other hand, the functional role of some of them remains unclear, especially in the context of the blood microbiome of healthy individuals. Although there is a point of view that there are no absolutely healthy people, the genomes of individuals are noted to be carriers of various potential pathological structures. Note that the authors used adequate methods for both obtaining data sets and bioinformatic and statistical processing to prove their hypothesis. In general, this manuscript is recommended for publication. I offer several technical recommendations:

1. The authors did not provide additional tables; therefore, their quality and necessity were not assessed.

2. Methods, Datasets section, 1 paragraph: It is necessary to provide a reference to the citation in the text: "In the original Natri et al. study…».

3. Methods, Datasets section, paragraph 2: It is necessary to agree with the list of references and correct the author’s last name “The first dataset comes from Tran et al. [28, 29] …”.

6. PLOS authors have the option to publish the peer review history of their article (what does this mean?). If published, this will include your full peer review and any attached files.

Reviewer #1: No

Reviewer #2: No

---

## [Author Response · Author response to Decision Letter 1]

2 May 2025

We have uploaded an updated document (response_to_editor_reviewers_PLOSONE_EiC_02052025.docx) addressing all the point-by-point comments from the editor and reviewers. The response is also detailed below:

1. Point-by-point description of the revisions

Editor:

a) Please explain how the author relates this study's impact to John P. Hanley et al. (2021) and Louis et al. (2005). What is the significance of whole blood transcriptome analysis, which identified distinct gene sets that correlated with viremia and revealed varying gene expression patterns before and after infection, implying immune state changes?

Our response: We are not sure which exact papers the editor referred to, but the comment is not within the immediate scope of our manuscript as our study contains only pre-infectious individuals.

b) What are the hematological changes and cellular composition changes and Gene Expression Patterns in relational to host and pathogen?

Our response: Our previous works with this dataset have covered this, and we have cited them (references 25-27; https://journals.plos.org/plosgenetics/article?id=10.1371/journal.pgen.1008749, https://www.sciencedirect.com/science/article/pii/S0002929721004377, https://www.biorxiv.org/content/10.1101/2021.01.07.425684v1). In short, we found no significant difference in gene expression levels or cellular composition associated with pathogen loads in our cohort. This is not surprising as our cohort is comprised of self-reported healthy individuals, so even those with detectable malaria loads were not presenting symptoms.

c) Why did the author choose an RNA-based investigation while prior research using mNGS and shotgun metagenomics demonstrated a more extensive analysis?

Our response: We have mentioned in our abstract and discussion sections how this approach is not a one-to-one replacement for shotgun metagenomics as a main surveillance tool, but rather a valuable retrospective tool to look at transcriptomic data generated in the past with no matched metagenomic data available.

d) Author mentioned “Recent studies have found no evidence of a core microbiome circulating in the blood of healthy Stanley plot individuals” and “However, not enough studies have been done to confirm whether this lack of a core microbiome is also consistent in regions where infectious diseases are endemic” but plethora of literature is available depicting role of core microbiome circulating in the blood of healthy individuals –

e) Study by Païssé et al. (2016), Study by Castillo et al. (2019), Blauwkamp et al….. so on, Nature study by Ling et al. (2023) - large-scale study analyzed blood samples from 9,770 healthy individuals.

Our response: The consensus in the field is clearly converging on there being no core blood microbiome. We believe the Ling et al study mentioned by the editor is actually a reference to Tan et al. (2023) (https://www.nature.com/articles/s41564-023-01350-w), which emphasize the notion that the blood of healthy individuals do not support a consistent core microbial community. We cited this paper in our manuscript as reference number 19 (page 14, first paragraph of discussion in our manuscript) as it supports our argument. We also cited Castillo et al. (2019) as reference number 48, in which the paper presents the evidence of eukaryotes and viruses in blood transcriptome yet also challenges their presence in healthy human blood, not concluding their definite existence and encouraging researchers to be more careful with experimental design (which also supports our main argument).

This topic in the microbiome / metagenomics field has been, and remains highly debated, with new evidence challenging the accepted paradigm; the Païssé et al. (2016) reference the editor mentioned is already 8 years old and more recent and robust evidence (like the Tan et al. paper) have come out.

We’re unsure which Blauwkamp et al. paper the editor referred to as he did not specify the year, but the one in 2019 (https://www.nature.com/articles/s41564-018-0349-6) is not relevant to the author’s point in depicting circulating microbiome in healthy individuals.

f) Why does the author focus exclusively on Indonesia—specifically Mentawai, Sumba, and the Indonesian portion of New Guinea Island—as the data set for this study?

Our response: We have mentioned in the abstract and introduction how Indonesia is such an underrepresented population with high numbers of endemic and emerging infectious diseases, and why we specifically chose those three regions (These populations span a gradient from west to east across Indonesia, thus capturing pathogens along the main geographical axis of the country). There seems to be a fundamental misunderstanding of our manuscript by the editor, and a re-read or a different editor may help clarify this point.

g) Enhance the methodology section by providing a detailed description that emphasizes sample collection, RNA isolation procedures, library preparation, quantification, and host DNA depletion techniques. Include the catalogue information for the kits used to facilitate replication by others, as well as the parameters employed for the Illumina HiSeq 2500, and ensure appropriate citations are included.

Our response: This was already presented and provided in the original paper (which we’ve cited). It is not a new dataset and therefore would be weird to provide them again here.

h) Provide the consent information for all human individuals (Weight, Hight, BMI etc).

Our response: Consent is not related to anthropometric descriptors. Regardless, we’ve always cited the original paper the data was first presented in.

i) Provide the mapped and unmapped paired-end data separately?

Our response: This was already presented and provided in the original paper (which we’ve cited). It is not a new dataset and therefore would be weird to provide them again here.

j) Please verify that the given data files encompass all individual samples in the repository.

Our response: This was already presented and provided in the original paper (which we’ve cited). It is not a new dataset and therefore would be unfit to provide them again here.

k) Instead of a circular bar plot, the author should present a stanley plot that shows the interconnected changes between the samples; this will provide a better understanding of the data.

Our response: The editor is very unclear in their demands. There is no such widely acceptable type of plotting style as a “Stanley plot”. We simply do not understand what the editor meant by this. If the editor invented this type of plots, we are not aware.

l) For compositional data analysis, the author must execute isometric log-ratio (ILR) transformation, which preserves all metric aspects of the data, coordinates with a non-singular covariance matrix, and allows for simpler interpretation of results. Alternatively, utilise the Robust Centred Log-Ratio (rCLR) and Inter-Quartile Log-Ratio (IQLR) Transformations.

Our response: We have used the Centred Log-Ratio (CLR) transformation for our data, which is adequate for compositional data, as we have explained in the methods section (page 7, Sample clustering subsection).

m) The justification for doing Welch's t-test in this investigation is unclear author should use Limma-Voom, NOIseq, SAMseq, etc. for analysis?

Our response: The methods the editor suggested are not recommended for microbiome data. As explained in our method section, we used ALDEx2 to test for differences in species composition between populations, which applies CLR-transformation to correct for uneven library depth and data compositionality and uses Welch’s t-test for significance test (page 7, Differential abundance testing and diversity estimation subsection). We ran Welch’s t-test and default 128 Monte Carlo simulations for the phylum-level analyses, as ALDEx2 is more suited for lower-level classification (species, genus) differential abundance.

n) In discussion part author mentioned “Our understanding of pathogens found within remote regions of Indonesia, along with their impact on gene expression, is limited. But several published studies on global scenario showed impact of pathogens are associated with environmental factors like temperature, humidity, location of longitude Natri et al., 2020 etc. As author used European data in this study but some studies mentioned the commonalities in gene regulation between Indonesia and Europe (Natri wt al., 2022), So, instead of using European data, I strongly recommend that the author conduct a comparative and conclusive study using data from other continents.

Our response: We also included an African cohort (Mali) for validation, which resembles the endemic / rural environment of our Korowai donors more. Again, there seems to be a fundamental misunderstanding of our manuscript by the editor, and a re-read or a different editor may help clarify this point.

Reviewer 1:

The manuscript titled “The Whole Blood Microbiome of Indonesians Reveals Translocated and Pathogenic Microbiota” presents the use of transcriptomics data to explore the presence of pathogens in whole blood samples. The authors have responded to previous reviewer questions; however, a few minor comments are provided below:

1. If the authors agree, the title can be revised to "Whole Blood Transcriptomics Analysis of Indonesians…" to provide a clearer picture of the study.

Our response: We thank Reviewer 1 for the suggestion and have amended it.

2. If the primary aim of the study is to detect pathogens, a microbial-targeted metagenomic study could be considered. This approach may help address the issue of contaminated human reads that the authors encountered. Are there any previous studies that used metagenomic analysis for pathogen detection, and did they produce similar or different results? If using RNA-seq is more critical than metagenomics in this context, the authors could clarify this in the introduction or discussion section.

Our response: As mentioned in our abstract and discussion sections, our approach is more suited as a retrospective surveillance tool in cases where only past transcriptomic data are available with no matched metagenomic data or fresh samples, not as a main disease surveillance tool. We have cited a few studies using microbial-targeted metagenomic approaches for pathogens and mentioned them in the introduction, such as the study by Kafetzopoulou et al. (2019) who used Nanopore sequencing for Lassa virus (https://pmc.ncbi.nlm.nih.gov/articles/PMC6855379/).

Reviewer 2:

The manuscript under review focuses on a relevant but controversial topic: Assessing the state of the blood microbiome. I would like to focus on the ambiguity of the task that the authors set in their study. On the one hand, the presence of genome fragments of certain types of microorganisms in the blood was confirmed by experimental studies. On the other hand, the functional role of some of them remains unclear, especially in the context of the blood microbiome of healthy individuals. Although there is a point of view that there are no absolutely healthy people, the genomes of individuals are noted to be carriers of various potential pathological structures. Note that the authors used adequate methods for both obtaining data sets and bioinformatic and statistical processing to prove their hypothesis. In general, this manuscript is recommended for publication. I offer several technical recommendations:

Our response: We thank Reviewer 2 for their fair assessment.

1. The authors did not provide additional tables; therefore, their quality and necessity were not assessed.

Our response: We have provided 8 supplementary tables containing the read depth quality and differential abundance results of all the involved datasets. We understand if Reviewer 2 might not have received the supplementary files during the review process.

2. Methods, Datasets section, 1 paragraph: It is necessary to provide a reference to the citation in the text: "In the original Natri et al. study…».

Our response: We thank Reviewer 2 for pointing this out and have amended it.

3. Methods, Datasets section, paragraph 2: It is necessary to agree with the list of references and correct the author’s last name “The first dataset comes from Tran et al. [28, 29] …”.

Our response: We thank Reviewer 2 for pointing this out and have confirmed the author’s name is correct (Tuan M. Tran, hence Tran et al.).

---

## [Editor Report · Decision Letter 1]

27 Jun 2025

PONE-D-24-24494R1Whole blood transcriptomics analysis of Indonesians reveals translocated and pathogenic microbiota in bloodPLOS ONE

Dear Dr. Gallego Romero,

Thank you for submitting your manuscript to PLOS ONE. After careful consideration, we feel that it has merit but does not fully meet PLOS ONE’s publication criteria as it currently stands. Therefore, we invite you to submit a revised version of the manuscript that addresses the points raised during the review process.

Please find my comments as new editor/ reviwer 3 below.

We look forward to receiving your revised manuscript.

Kind regards,

Sylvia Maria Bruisten, Ph.D

Academic Editor

PLOS ONE

Journal Requirements:

Additional Editor Comments:

This manuscript was previously commented on by another editor and two reviewers. The two reviewers had minor comments which were well answered in this first Rebuttal version. The comments from the editor met with difficulties for the authors to answer since there seem to be misunderstandings on what this manuscript aims to describe. As the new scientific editor I think that this is a well written and mostly comprehensive manuscript that describes interesting data on transcriptome (RNA seq) data on previously collected whole blood samples from 3 Indonesian populations. These data were compared to similar data from a population in Mali and a small population from London, UK.

I have one major and some minor comments to further improve the manuscript.

Major comment

As was also mentioned earlier by reviewer 2, Supplementary Tables 1 to 8 cannot be found in the document submitted to PlosOne (PONE-D-24-24494R1). Please provide these Suplementary Tables with the next Rebuttal version.

Minor comments

Abstract

It is mentioned that ‘two pathogens—Flaviviridae and Plasmodium—are the most predominantly abundant’. I suggest to change ‘pathogens’ into ‘taxa’ since that is more accurate. If the authors like they can add that these taxa includes pathogens such as Dengue viruses and malaria causing parasites.

In addition there are 5 ‘key highlights’ mentioned in the letter to the editor from May 2025. The first three key points deserve to be also mentioned in the abstract because in my view this is what the reader wants to know. There is no limitation in the number of words for an abstract for PlosOne, so this can be formulated it as comprehensive as needed.

Results

Supplemental Figure 1 and Figure 2: The legend colours and text do not everywhere line up; please adjust so these are aligned properly. For example ‘Chordata’ in Suppl Fig 1 is now next to a blue block whereas it should be next to a purple one. And ‘KOR’ should be next to the red blocks.

Supplementary Figure 5 contains interesting data which I think deserve to be shown in the main text. Please consider moving it there.

Reviewers' comments: No extra comments, reviewers were not again invited since their pints were properly addressed.

---

## [Author Response · Author response to Decision Letter 2]

2 Jul 2025

Major comment

As was also mentioned earlier by reviewer 2, Supplementary Tables 1 to 8 cannot be found in the document submitted to PlosOne (PONE-D-24-24494R1). Please provide these Suplementary Tables with the next Rebuttal version.

Our answer: We thank the editor for pointing this out and have now included supplementary tables 1-8 in the new version (as two separate excel documents).

Minor comments

Abstract

It is mentioned that ‘two pathogens—Flaviviridae and Plasmodium—are the most predominantly abundant’. I suggest to change ‘pathogens’ into ‘taxa’ since that is more accurate. If the authors like they can add that these taxa includes pathogens such as Dengue viruses and malaria causing parasites.

Our answer: We thank the editor for the suggestion and have amended the section.

In addition there are 5 ‘key highlights’ mentioned in the letter to the editor from May 2025. The first three key points deserve to be also mentioned in the abstract because in my view this is what the reader wants to know. There is no limitation in the number of words for an abstract for PlosOne, so this can be formulated it as comprehensive as needed.

Our answer: We thank the editor for the suggestion and have incorporated the first three key points into the abstract.

Results

Supplemental Figure 1 and Figure 2: The legend colours and text do not everywhere line up; please adjust so these are aligned properly. For example ‘Chordata’ in Suppl Fig 1 is now next to a blue block whereas it should be next to a purple one. And ‘KOR’ should be next to the red blocks.

Our answer: We thank the editor for the observation. It seems that the tracked conversion from our Latex pdf output to a word document has caused the individual elements of the figures to be individualized instead of merged as one, as the figures look fine on pdf. We have now amended this.

Supplementary Figure 5 contains interesting data which I think deserve to be shown in the main text. Please consider moving it there.

Our answer: We thank the editor for the suggestion and have moved Supplementary Figure 5 into the main text as Figure 3.

---

## [Editor Report · Decision Letter 2]

8 Jul 2025

Whole blood transcriptomics analysis of Indonesians reveals translocated and pathogenic microbiota in blood

PONE-D-24-24494R2

Dear Dr. Gallego Romero,

We’re pleased to inform you that your manuscript has been judged scientifically suitable for publication and will be formally accepted for publication once it meets all outstanding technical requirements.

Kind regards,

Sylvia Maria Bruisten, Ph.D

Academic Editor

PLOS ONE

Additional Editor Comments (optional):

All comments were taken into consideration and all points were adjusted to satisfaction.
---

## [Editor Report · Acceptance letter]

PONE-D-24-24494R2

PLOS ONE

Dear Dr. Gallego Romero,

I'm pleased to inform you that your manuscript has been deemed suitable for publication in PLOS ONE. Congratulations! Your manuscript is now being handed over to our production team.

Kind regards,

on behalf of

Dr. Sylvia Maria Bruisten

Academic Editor

PLOS ONE